# Prostate Cancer Survivors Present Long-Term, Residual Systemic Immune Alterations

**DOI:** 10.3390/cancers14133058

**Published:** 2022-06-22

**Authors:** Katalin Balázs, Zsuzsa S. Kocsis, Péter Ágoston, Kliton Jorgo, László Gesztesi, Gyöngyi Farkas, Gábor Székely, Zoltán Takácsi-Nagy, Csaba Polgár, Géza Sáfrány, Zsolt Jurányi, Katalin Lumniczky

**Affiliations:** 1National Public Health Center, Unit of Radiation Medicine, Department of Radiobiology and Radiohygiene, 1221 Budapest, Hungary; balazs.katalin@osski.hu (K.B.); safrany.geza@osski.hu (G.S.); 2Doctoral School of Pathological Sciences, Semmelweis University, 1085 Budapest, Hungary; 3Department of Radiobiology and Diagnostic Onco-Cytogenetics and The National Tumorbiology Laboratory, Centre of Radiotherapy, National Institute of Oncology, 1122 Budapest, Hungary; kocsis.zsuzsa@oncol.hu (Z.S.K.); farkas.gyongyi@oncol.hu (G.F.); szekely.gabor@oncol.hu (G.S.); juranyi.zsolt@oncol.hu (Z.J.); 4Centre of Radiotherapy and The National Tumorbiology Laboratory, National Institute of Oncology, 1122 Budapest, Hungary; agoston.peter@oncol.hu (P.Á.); jorgo.kliton@oncol.hu (K.J.); gesztesi.laszlo@oncol.hu (L.G.); takacsi.zoltan@oncol.hu (Z.T.-N.); polgar@oncol.hu (C.P.); 5Department of Oncology, Semmelweis University, 1122 Budapest, Hungary

**Keywords:** prostate cancer, radiotherapy, immune phenotyping, adaptive immune response, innate immune response

## Abstract

**Simple Summary:**

The development of cancer is very often accompanied by systemic immune alterations which can be further aggravated by major anti-cancer therapies. However, there is very little known about how long these alterations persist in patients successfully cured of cancer. The aim of our work was to investigate how cancer and radiotherapy as major anti-cancer treatment modalities impact the immune system long after the successful treatment of a tumor. We investigated prostate cancer patients treated with a special form of radiotherapy (low-dose rate brachytherapy) often used for the treatment of prostate cancer and followed a wide range of immune parameters at regular intervals up to 3 years after the start of the treatment. Our results showed that some immune alterations did not recover after the treatment of the disease, on the contrary, they persisted, and in some cases got even worse. Further studies are needed to explain the causes and the potential long-term consequences of these alterations.

**Abstract:**

Background: The development of cancer and anti-tumor therapies can lead to systemic immune alterations but little is known about how long immune dysfunction persists in cancer survivors. Methods: We followed changes in the cellular immune parameters of prostate cancer patients with good prognostic criteria treated with low dose rate brachytherapy before and up to 3 years after the initiation of therapy. Results: Patients before therapy had a reduced CD4+ T cell pool and increased regulatory T cell fraction and these alterations persisted or got amplified during the 36-month follow-up. A significant decrease in the total NK cell number and a redistribution of the circulating NK cells in favor of a less functional anergic subpopulation was seen in patients before therapy but tumor regression led to the regeneration of the NK cell pool and functional integrity. The fraction of lymphoid DCs was increased in patients both before therapy and throughout the whole follow-up. Increased PDGF-AA, BB, CCL5 and CXCL5 levels were measured in patients before treatment but protein levels rapidly normalized. Conclusions: while NK cell dysfunction recovered, long-term, residual alterations persisted in the adaptive and partly in the innate immune system.

## 1. Introduction

The development and progression of malignant tumors may strongly influence systemic immune parameters in cancer patients; anti-cancer treatment modalities, such as RT or chemotherapy have immunomodulatory effects. It has been demonstrated in several experimental and clinical studies that tumor progression is often accompanied by systemic immune suppression [1,2,3,4,5,6,7]. 

According to the 2020 cancer statistics of the International Agency for Research on Cancer (IARC), prostate cancer is the third most common cancer type (7.1% of the 19.3 million total cases) after lung cancer and female breast cancer [8]. Prostate specific antigen (PSA) is currently the most widespread and accepted biomarker for prostate cancer monitoring; however, it is not an ideal tumor marker. Over the past decade, liquid biopsy investigations have received increasing attention to complement the risk group classification of prostate cancer patients [9].

Radiotherapy (RT) is one of the main treatment methods for organ confined and locally advanced prostate cancer along with hormone therapy and surgery. Combined therapeutic regimens are often used in high risk or locally advanced cases, while chemo- and targeted therapy are also applied for advanced, metastatic prostate cancer [10].

Regarding prostate cancer RT, different techniques and fractionation schedules are routinely applied with comparable efficiency in terms of cancer cure, though the kinetics of radiation energy deposition, the volume of irradiated surrounding healthy tissue, and therefore, the rate and severity of side effects are largely different. Interstitial prostate brachytherapy is a type of RT, where radiation sources are implanted into the prostate. Low dose rate (LDR) brachytherapy is a subtype of brachytherapy performed by the permanent insertion of capsulated radioactive isotopes mainly with low-energy photon-emitting sources (commonly ^125^I or ^103^Pd) [11]. Its first use dates back to 1914 [12], and the technique later evolved to be a safe and effective treatment modality for prostate cancer [13]. LDR brachytherapy is characterized by a low and continuous energy deposition locally in the tumor over a period of several months. It can be regarded as a source of chronic irradiation with regulated energy deposition restricted within the tumor. Irradiation of surrounding healthy tissues is limited; therefore, side effects are also usually milder than after other RT techniques. 

RT can influence the immune status of patients. Several papers investigated the influence of RT on the intratumoral immune response in various tumor types and how RT-induced changes to local immune parameters impact tumor behavior and long-term response to therapy [14,15]. On the other hand, local RT can modulate systemic immune responses as well [16,17]. While formerly RT was considered a therapeutic agent leading to or aggravating immune suppression in cancer patients, recently it has been shown to stimulate certain elements of the antitumor immune response [18,19,20,21,22,23,24].

Changes in the systemic immune response have important consequences for tumor metastasis, therapy-related late tissue reactions, and the development of second malignant neoplasms and can help with the individualization of therapy and the identification of patient subgroups who could benefit from certain therapeutic combinations. A major benefit of a better understanding of RT-related systemic immune changes would be the identification of immune-related liquid biopsy markers of therapy response and long-term prognosis. Relatively few studies investigated RT-induced systemic immune changes in cancer patients and the long-term follow-up of these parameters is lacking. Yoon et al. investigated the systemic changes in the neutrophil-to-lymphocyte ratio (NLR) caused by RT in patients with breast cancer and found that patients with an NLR higher than 3.49 after RT showed a significantly higher recurrence rate compared to those with a low NLR, therefore, NLR can serve as a negative prognostic factor [25]. Dovsak et al. showed that lymphocyte numbers, including B helper and cytotoxic T lymphocytes in patients with oral cancer treated with RT, remained under pre-treatment levels 1 year after treatment, indicating that RT induced long-lasting systemic immunological changes [26]. It was shown in both pre-clinical studies and in melanoma patients that local RT enhanced systemic responses to anti-cytotoxic T-lymphocyte-associated protein 4 (CTLA4) immunotherapy [27,28]. We previously investigated RT-induced changes in systemic immune and inflammation parameters of head and neck squamous cell carcinoma patients and found that RT intensified the systemic immune suppression present in cancer patients before irradiation [29]. 

In this study, we investigate the long-lasting changes in systemic immune parameters in prostate cancer patients treated with low dose rate brachytherapy. Patients were followed up for 36 months after treatment initiation and the kinetics of changes in the fraction of various lymphocyte subpopulations and in selected soluble plasma proteins were determined at regular intervals. 

## 2. Materials and Methods

### 2.1. Patient Enrolment and Follow-Up

Twenty-one prostate cancer patients aged 52–77 years (median 70 years) treated with low-dose rate brachytherapy and 36 age-matched healthy controls were recruited at the National Institute of Oncology (NIO), Budapest, Hungary based on ethical permissions released by Scientific and Research Ethics Committee of the Medical Research Council (permit numbers 16738-2/2015, 20042-4/2016, 20542-2/2019). The research was conducted in accordance with the principles of the Declaration of Helsinki as well as with relevant Hungarian and EU regulations. Informed consent was obtained from all human subjects enrolled in the study. Low (*n* = 7) and intermediate (*n* = 14) risk patients based on TNM, Gleason grade of the tumors and PSA according to the risk classification by d’Amico et al. [30] were included in the study. If patients developed lymph node or distant metastases during the follow-up period, they were excluded from the study (patients p072, p085, p091). 

Patients were treated with LDR brachytherapy; the applied radiation source was an ^125^I isotope with a half-life of 59.49 days. The initial dose rate was 1.692 Gy/day and a 145 Gy total dose was delivered continuously to the prostate gland for approximately 12 months (Appendix A). The radiation source was implanted permanently in the prostate gland. Thirteen patients received hormonal therapy as well prior to radiotherapy, but none were treated by surgery or chemotherapy. Hormone therapy was stopped after implantation and consisted of either luteinizing hormone-releasing hormone (LHRH) agonist (*n* = 10) or LHRH antagonist (*n* = 2) or total androgen blockade (TAB) therapy (*n* = 1).

Patients were followed at regular intervals (before as well as 3, 6, 12, 24 and 36 months after seed implantation), and blood for immunological analysis was collected at these time points. The International Prostate Symptom Score (IPSS) and Quality of Life (QoL) tests were used to survey the severity of LDR brachytherapy-induced side effects based on the cancer patients’ subjective evaluation [31,32,33]. Acute and chronic genitourinary (GU) and gastrointestinal (GI) side effects were registered according to toxicity criteria of the Radiation Therapy Oncology Group (RTOG) and the European Organization for Research and Treatment of Cancer (EORTC). Side effects were scored and blood was collected at each follow-up occasion. Collected blood was used for PSA measurements as well as peripheral blood mononuclear cell (PBMC) and plasma isolation for further analyses.

### 2.2. Blood Collection, Isolation of PBMCs and Plasma

Blood samples were collected at NIO on every follow-up occasion. Control blood samples were collected from age-matched healthy volunteers.

Approximately 10 mL blood/patient/sampling time was collected in heparinized tubes (Vacuette tube, Greiner Bio-One Gmbh, 4550 Kremsmünster, Austria) and diluted with phosphate buffered saline (PBS) in a 1:1 ratio. Six milliliters of diluted blood were pipetted on 2 mL Histopaque (Sigma-Aldrich, Co., 3050 St. Louis, MO, USA) and centrifuged (Thermo Fisher Scientific, Waltham, MA, USA) for 15 min at 2000 rpm at room temperature in zero fall-off mode. PBMCs were isolated from the layer under the plasma and above the Histopaque. Cells were washed twice in PBS. PBMCs were aliquoted in RPMI-1640 containing 10% dimethyl sulfoxide and 50% fetal bovine serum at approx. 1.5 × 10^6^ cells/mL, gradually cooled down to −80 °C using a NALGENE™ Cryo 1 °C Freezing Container device filled with 2-propanol puriss (Reanal Labor, Budapest, Hungary) for 24 h in CryoPure tubes (Sarstedt, Nümbrecht, Germany). The supernatant (plasma) was centrifuged (10 min, 2500 rpm, 25 °C) to remove residual cellular debris. Frozen PBMCs and plasma were transferred to National Public Health Centre in dry ice and PBMCs were further stored in liquid nitrogen, while plasma was stored at −80 °C until assayed. 

### 2.3. Immune Phenotyping of PBMCs

Frozen PBMC samples were thawed in a 37 °C water bath and the number of viable cells was determined by trypan blue staining. One million cells were suspended in 100 µL staining buffer (consisting of Hank’s balanced salt solution containing 1% bovine serum albumin) in 5 mL centrifuge tubes (Sarstedt, Nümbrecht, Germany). For cell surface staining, cells were incubated with the corresponding antibodies at 4 °C in dark for 30 min. For intracellular staining, cells were fixed and permeabilized using Biolegend Foxp3 Fix/Perm Buffer set (BioLegend, San Diego, CA, USA) according to the manufacturer’s instructions, followed by the addition of the corresponding antibodies. Stained cells were stored in 1% paraformaldehyde until analysis but not longer than 24 h. The following directly labeled anti-human monoclonal antibodies were used for phenotypical analysis: CD4-APC, CTLA4-PE, PD1-PerCP/Cy5.5, Foxp3−Alexa Fluor488, CD39-PerCP/Cy5.5 and Ki-67-PE for CD4+ T and regulatory T cells (Tregs); CD11b-FITC, CD33-PerCP/Cy5.5 and HLA-DR-APC for myeloid-derived suppressor cells (MDSCs); CD16-FITC, CD56-PE and CD3-APC for natural killer cells (NKs) and Lineage1 cocktail (CD3, CD14, CD16, CD19, CD20, CD56)-FITC, HLA-DR-PerCP, CD123-PE and CD11c-APC for dendritic cells (DCs). All antibodies were purchased from BioLegend.

The samples from multiple time points of each individual patient were investigated at the same time, in one flow cytometry assay.

### 2.4. Hierarchical Gating Strategy

PBMC subpopulations were analyzed and quantified by flow cytometry (FACS Calibur, Becton Dickinson, CA, USA). A hierarchical gating strategy was created for effective flow cytometry data analysis using the Kaluza 1.5a software (Beckman Coulter, CA, USA) as previously described [29]. Shortly, CD4+ cells (including Tregs) and NK cells were analyzed within the lymphocyte gate. Tregs were quantified as CD4+Foxp3+ cells. CTLA4+ cell subpopulations were evaluated within the whole CD4+ gate as well as within the individual Treg gate (CD4+Foxp3+) and effector T cells (Teff) gate (CD4+Foxp3−). PD1+ and CD39+ cells were evaluated within the CD4+ gate in a similar manner to CTLA4. The proliferating index of CD4+ cells was determined by evaluating the fraction of Ki67+ cells, applying the same gating strategy as for CTLA4 and PD1. NK cells were identified as CD3-CD16+CD56+ cells within the lymphocyte gate. Five different maturation states of circulating NK cells were distinguished based on the expression level of the adhesion molecule CD56 and the activating receptor CD16 within the CD3- lymphocyte gate (Appendix A). These are immature precursor NK cells (CD56^bright^CD16-), immature/early mature (CD56^bright^CD16^dim^), mature cytotoxic (CD56^dim^CD16^bright^), degranulating (CD56^dim^CD16-) and anergic NK cells (CD56-CD16^bright^). Natural killer T (NKT)-like cells were investigated as CD3+CD56+ cells within the lymphocyte gate (Appendix A). DCs were analyzed within HLA-DR+Lineage1- PBMCs. Myeloid DCs were considered CD11c+Lineage1- cells, while lymphoid or plasmacytoid DCs were considered CD123+Lineage-1- cells. MDSCs were identified as HLA-DR-CD11b+CD33+ cells in the total PBMCs excluding the lymphocyte gate. For most cell subpopulations the relative proportion (%) of the population was determined. However, for expression of the CTLA-4, PD-1 and CD-39 activation markers, the median fluorescent intensity was determined as well in order to evaluate the level of marker expression. 

### 2.5. Investigating the Protein Profile of Plasma Samples

The protein profiles of the plasma samples, including the growth factor and proinflammatory chemokine panels, were analyzed using the high throughput multi-analyte flow assay, using the LEGENDplex kit (BioLegend, San Diego, CA, USA), which allows the simultaneous quantification of 13 proteins within one full panel (Appendix A). 

Frozen plasma samples were thawed at room temperature and centrifuged (2500 rpm, 25 °C, 8 min) to remove particulates and debris. Plasma samples were diluted with assay buffer (provided by the manufacturer) two-fold in the case of the human growth factor panel and 20-fold in the case of the human proinflammatory chemokine panel. Reagent and standard preparations and the assay procedure were made according to the manufacturer’s instructions. Samples were read on a flow cytometer (FACS Calibur, Becton Dickinson, CA, USA) on the same day of the assay. The Cell-Quest™ Pro data acquisition and analysis software version 4.0.2 was used for data analysis (Becton Dickinson). 

### 2.6. Statistical Analyses

Statistical analyses were performed with the GraphPad Prism version 6.00 for Windows software (GraphPad Software, La Jolla, CA, USA). Data are illustrated as scatter dot plots, and data are shown as individual data points together with the mean and standard deviation (SD). Two-tailed unpaired *t*-test was used to test differences between healthy controls and cancer patients at different time points. *p*-values lower than 0.05 were considered statistically significant using a 95% confidence interval (CI). Correlation analyses were executed with Pearson correlations.

## 3. Results

### 3.1. Demographics and Clinical Parameters of Prostate Cancer Patients Treated with LDR Brachytherapy

All patients had tumors in the T1 or T2 stage at the start of RT (24% in T1c, 33% in T2a, 14% in T2b and 29% in T2c) (Table 1). The PSA level before the start of RT was within the normal range (at NIO values below 3 ng/mL are considered normal) for 43% of the patients, while 29% had PSA levels between 3 and 10 ng/mL and 19% had PSA levels above 10 ng/mL. For all but two patients, abnormal PSA levels returned to normal within three months after the implantation of the radiation source. For patient P058 this happened within 6 months and for patient P085 within 9 months (Table 2). Based on the TNM classification, initial PSA levels and Gleason Score, patients were categorized as low risk (risk classification 1) (33%) and intermediate risk (risk classification 2) (67%) (Table 1). All patients were treated with LDR brachytherapy as described in the materials and methods section. More than half of the patients (62%) were treated with hormone therapy before the implantation. Eight patients (38%) did not receive any hormone therapy.

The majority of patients (76%) had grade 2 cumulative late GU side effects, and just one patient had more severe late GU symptoms. According to the pre-treatment IPSS scores 14% (3/21) of the patients had mild (score 1–7), 43% (9/21) had moderate (score 8–19) and 43% (9/21) had severe (score 20–35) urination symptoms. In contrast, the GI side effects were milder; 71% of patients had no GI side effects, 24% of them had grade 1 and 5% of patients had symptoms of grade 2 severity. The test of QoL showed that 19% (4/21) of patients were pleased, 34% (7/21) were mostly satisfied, 19% (4/21) were mixed, 19% (4/21) were mostly dissatisfied and 10% (2/21) were unhappy with the quality of their life due to urinary symptoms. No patients with grade 4 GU or GI side effects and with a 6 QoL score were identified. (Appendix A). 

### 3.2. Low Dose Rate Brachytherapy Induces Persistent Changes in the Distribution and Phenotype of Several PBMC Subpopulations in Prostate Cancer Patients

Therapy-related changes in the cellular immune parameters of low-dose rate brachytherapy-treated prostate cancer patients were investigated by immune phenotyping of the PBMCs before therapy and 3, 6, 12, 24, 36 months after seed implantation and were compared to the immune phenotype of healthy controls. For certain patients not all investigated immune cell phenotypes are available for all interim follow-up periods due to the insufficient PBMC yield in some of the collected blood samples. 

The fraction of CD4+ T cells within the lymphocyte population of prostate cancer patients was moderately lower than in healthy controls throughout the whole follow-up but interindividual variation was high. A statistically significant difference was only seen in prostate cancer patients 6 and 24 months after implantation of the radiation source (Figure 1). 

The fraction of CD4+Foxp3+ Tregs in the lymphocyte population of healthy controls was on average 0.52% (95% CI: 0.18–0.85%). The fraction of Tregs in prostate cancer patients before therapy and up to 24 months after implantation did not differ significantly from that of healthy controls. However, 36 months after treatment, the Treg fraction increased to 1.15% (95% CI: 0.88–1.42%), which represents a significant increase compared to both healthy control and pre-treatment values (Figure 2 and Appendix A). 

Next, we investigated two activation markers on CD4+ T cells, with their expression indicating the acquisition of immune inhibitory (CTLA4) [34,35] or immune tolerant, exhausted (PD1) phenotypes [36]. The fraction of CD4+ T cells expressing CTLA4 in cancer patients before seed implantation (3.01% mean with 1.38–4.63% 95% CI) was slightly higher but not significantly different from healthy controls (2.24% mean with 1.27–3.22% 95% CI), and a mild decrease was noticed after treatment, reaching significantly lower CTLA4+ T cell fractions 36 months after implantation (1.32% mean with 0.95–1.69% 95% CI) (Figure 3A). This was due to changes in the fraction of CTLA4 expressing Teff cells since this cell subpopulation decreased significantly at late time points after seed implantation (24 months: 1.25% mean with 0.96–1.54% 95% CI and 36 months: 0.71% mean with 0.49–0.93% 95% CI) compared to both healthy controls (1.96% mean with 1.02–2.91% 95% CI) and prostate cancer patients before seed implantation (2.53% mean with 1.09–3.96% 95% CI) (Figure 3B). No significant changes were detected in the amount of CTLA4+ cells within the CD4+Foxp3+ lymphocytes (Figure 3C). On the other hand, the analysis of the distribution of Foxp3+ and Foxp3− cells within the CTLA4+ CD4+ lymphocytes showed that the fraction of Foxp3+ cells was 1.68-fold higher in prostate cancer patients before seed implantation compared to healthy controls (20.15% mean with 10.84–29.46% 95% CI in cancer patients before RT, 12% mean with 4.98–19.02% 95% CI in healthy controls), and this shift towards Foxp3+ cells continued to increase during the 36 months follow-up (33.91% mean with 24.97–42.85% 95% CI and 40.96% mean with 18.28–63.64% 95% CI in prostate cancer patients 24 months and 36 months after seed implantation, respectively) (Figure 3D). Moreover, the level of CTLA-4 expression on Foxp3+CD4+ Tregs (as evaluated by median fluorescence intensity) progressively increased in cancer patients after the initiation of radiotherapy, reaching significantly higher levels 24 and 36 months after seed implantation compared to both healthy controls and cancer patients before treatment (Figure 3E).

The fraction of PD1 expressing CD4+ T lymphocytes, CD4+Foxp3− Teff cells and CD4+Foxp3+ Tregs was not significantly different from healthy controls in cancer patients before or at various time points after brachytherapy (Figure 4A–C). A tendency for a redistribution similar to the CTLA4 was seen in the PD1+CD4+ T cell population in the favour of Foxp3+ cells, albeit changes were statistically not significant (Figure 4D). The expression level of PD-1 on Tregs, similarly to the expression of CTLA4 showed a mild but progressive increase in brachytherapy-treated prostate cancer patients, reaching significantly increased levels 36 months after seed implantation compared to patients before therapy (Figure 4E). 

A third functional marker, CD39 was also measured on CD4+ T cells. Neither the fraction of CD39 expressing Tregs nor CD39 expression level on Tregs was significantly different in cancer patients during the follow-up compared to before treatment (Appendix A). The fraction of CD39 expressing CD4+ Teff cells significantly increased in prostate cancer patients before therapy (3.32% mean with 1.32–5.32% 95% CI before therapy and 1.45% mean with 0.64–2.26% 95% CI in healthy controls) and levels further increased 3 months after seed implantation (5.33% mean with 2.37–8.30% 95% CI), followed by a slow decrease. The mildly increased CD39+CD4+ effector T cell fraction persisted even 3 years after therapy (2.66% mean with 1.97–3.35% 95% CI). The expression level of CD39 on CD4+ effector T cells was significantly higher after the 36-month follow-up of prostate cancer patients (Appendix A). 

The proliferation status of CD4+ T cells was also studied as a further indicator of their functional integrity and potential activation status. Proliferation status was investigated by measuring the expression of the Ki-67 proliferation marker. The proliferation of CD4+ T cells in prostate cancer patients before seed implantation was significantly higher than in healthy controls (3.80% mean with 1.61–6.00 95% CI proliferating CD4+ cells in cancer patients versus 1.36% mean with 0.78–1.93% 95% CI in healthy controls). This elevated CD4+ T cell proliferation persisted 3 months after the start of LDR brachytherapy, and progressively returned to control levels by month 12 after seed implantation but increased again at month 36, though it did not reach pre-treatment values (Figure 5A). After evaluating changes in the proliferative capacity of CD4+ Teff cells (Figure 5B) and Tregs (Figure 5C) separately, it could be seen that variations in the proliferative capacity of CD4+ T cells were attributable exclusively to CD4+ Teff cells since the proliferative capacity of Treg cells was not significantly changed throughout the follow-up period. It was interesting to see that the proliferative activity of Treg cells was higher than that of CD4+ effector T cells in healthy controls (6.18% mean with 3.60–8.75% 95% CI versus 1.58% mean with 0.68–2.47% 95% CI).

The level of circulating total NK cells (identified as CD16+CD56+CD3- lymphocytes) was significantly lower in prostate cancer patients before seed implantation compared to healthy controls (52.41% mean with 41.14–63.68% 95% CI and 67.82% mean with 62.02–73.62% 95% CI in patients before treatment and healthy controls, respectively). However, NK cell levels returned to control values as soon as 3 months after seed implantation (Figure 6A, Appendix A). No significant changes were detected in the fraction of immature precursor NK cells during the whole follow-up of prostate cancer patients compared to healthy controls (Figure 6B). Both the fraction of immature/early mature NK cells and that of mature, cytotoxic NK cells was significantly lower in prostate cancer patients before treatment compared to healthy controls (0.79% mean with 0.50–1.05% 95% CI and 45.99 mean with 34.88–57.10 95% CI in prostate cancer patients compared to 1.69% mean with 1.36–2.02 95% CI and 61.65 mean with 55.89–67.41 95% CI in healthy controls for immature/early mature NK cells and mature, cytotoxic NK cells, respectively) but returned to close to control values by month 6 after seed implantation (Figure 6C,D). Changes in the fraction of degranulating NK cells were mild and statistically not significant (Figure 6E). The fraction of anergic NK cells changed in the opposite direction to mature, cytotoxic and immature/early mature NK cells in cancer patients, showing a significant increase before seed implantation and 3 months after, while progressively decreasing at later time points and reaching 1.5-fold lower levels 36 months after therapy compared to healthy controls (6.00% mean with 4.35–7.60 95% CI in prostate cancer patients 36 months after RT and 9.01% mean with 7.77–10.25 95% CI in healthy controls) (Figure 6F).

Natural killer T-like cells were identified as CD3+CD56+ cells within the lymphocyte gate. The level of NKT-like cells significantly decreased in prostate cancer patients before the start of the therapy compared to healthy controls (4.90% mean with 2.53–7.27% 95% CI and 9.11% mean with 7.21–11.02% 95% CI in patients before RT and healthy controls, respectively) but quickly returned to control levels and remained unchanged during the follow-up period (Appendix A).

Next, we investigated systemic changes in MDSC levels in prostate cancer patients before and after RT. MDSCs were identified in the total PBMCs excluding the lymphocyte gate as CD11b+ and CD33+ double positive cells within the HLA-DR- cells. MDSC levels in healthy controls were 0.77% with 0.43–1.13% 95% CI. No significant differences in MDSC levels were noted in prostate cancer patients before seed implantation compared to healthy controls, while seed implantation induced a mild but significant reduction in the fraction of MDSCs starting from month 6 (0.42% mean with 0.06–0.77% 95% CI) and persisting even 36 months later (0.36% mean with 0.17–0.55% 95% CI) (Appendix A). 

Finally, two subpopulations of circulating DCs were studied: CD123+ plasmacytoid/lymphoid and CD11c+ myeloid DC. The level of lymphoid DCs significantly increased in prostate cancer patients and remained elevated throughout the follow-up period with moderate fluctuation at month 12 after RT (5.50% mean with 4.41–6.59% 95% CI, 16.05% mean with 0.70–31.40% 95% CI, 14.97% mean with 7.56–22.37% 95% CI, 15.55% mean with 10.45–20.65% 95% CI and 21.86% mean with 10.05–33.67% 95% CI in healthy controls, patients before therapy, at month 3, 6 and 24 after seed implantation, respectively) (Figure 7A). In contrast, the fraction of myeloid DCs was not significantly different from healthy controls in prostate cancer patients before and up to month 6 after seed implantation and mildly increased at later time points (19.53% mean with 16.22–22.85% 95% CI and 29.38% with 17.31–41.45% 95% CI in healthy controls and patients at month 24 after seed implantation, respectively) (Figure 7B). 

No correlations were found between cellular immunological parameters and either PSA levels or therapy-related side effects.

### 3.3. Increased Plasma Levels of Growth Factors and Chemokines in Prostate Cancer Patients Prior to RT

The long-term consequences of low-dose rate brachytherapy on plasma protein levels in prostate cancer patients in comparison to healthy controls were investigated by multi-analyte flow assay LEGENDplex kit at similar time intervals to immune cell phenotyping. 

Our analysis focused on growth factors reported to be involved in fibrosis and in endothelial dysfunction and on selected pro-inflammatory proteins and chemokines involved in systemic inflammatory processes. The majority of the investigated proteins were either undetectable (M-CSF, G-CSF, GM-CSF, TGF-α and IL-8; not shown) or did not change significantly in the plasma of prostate cancer patients during their follow-up (Appendix A). Two growth factors, PDGF-AA and PDGF-BB strongly increased in prostate cancer patients before therapy (3925 pg mean with 2367–5484 pg 95% CI and 13,332 pg mean with 8892–17,772 pg 95% CI in patients before treatment versus 1019 pg mean with 628.4–1410 pg 95% CI and 3865 pg mean with 2755–4974 pg 95% CI in healthy controls for PDGF-AA and PDGF-BB, respectively) (Figure 8A,B). Though, their levels rapidly normalized after treatment, reaching control values by month 6. Two chemokines, RANTES/CCL5 and ENA-78/CXCL5 showed similar, albeit milder dynamics to PDGF proteins in prostate cancer patients before treatment (3653 pg mean with 1068–6239 pg 95% CI and 325 pg mean with 211–439 pg 95% CI in patients before treatment versus 1339 pg mean with 1090–1589 pg 95% CI and 163 pg mean with 114–211 pg 95% CI in healthy controls for RANTES and ENA-78, respectively), and returned to control values by month 6 (Figure 8C,D). These data indicate that alterations in the level of the four investigated proteins were linked to the malignant status and tumor regression led to a quick normalization of plasma protein levels. Their levels did not correlate with pretreatment PSA values. 

## 4. Discussion

The aim of our study was to investigate long-term systemic immunological changes in cured prostate cancer patients treated with radiotherapy. Low- to intermediate risk patients were selected with T1 or T2 tumor stages, low pre-treatment PSA levels and low Gleason scores, with no lymph node or distant metastases at the time of enrolment, indicating a relatively homogeneous patient group with good prognostic criteria (Table 1). The majority of patients received hormone therapy, none were treated by surgery and all patients were treated with LDR brachytherapy. Therapy-related side effects were mild, most probably due to the applied RT protocol and in agreement with previous reports on LDR brachytherapy-induced late toxicities in prostate cancer patients [37,38,39,40]. Patient selection criteria allowed for the regular follow-up of the patients up to 36 months after the initiation of radiotherapy in order to investigate long-term phenotypical changes in several lymphocyte subpopulations known to be important players in the anti-tumor innate and adaptive immune response. 

Seed brachytherapy is usually applied for prostate cancer patients with a good prognosis. In the case of seed brachytherapy, the response rate evaluated right after the therapeutic intervention is not applicable, as the dose delivery is approximately one year long and the progression to relapse is slow in prostate cancer so we have to follow the patients for a minimum of 5 years, to gain a valid response rate. However, seed brachytherapy is known to result in excellent survival (15-year biochemical relapse-free survival was 80.4% in the study of Sylvester et al. and 85.9% in the study of D’Amico et al.) [30,41]. In our case, at five years, biochemical relapse-free survival is 89.9 ± 6.8%, local relapse-free survival is 86.8 ± 8.9%, and the overall survival is 90.5 ± 6.4%. Due to the very good response of patients to the therapy and the low relapse rates, it was not possible to compare immunological data with outcome variables.

Alteration in the CD4+ T cell pool was most probably related to the development of the malignant condition itself since prostate cancer patients had a mildly reduced CD4+ T cell pool which was not significantly influenced by LDR brachytherapy. Low CD4+ levels did not resolve after the treatment of the tumor itself, remaining below control values even after a 36-month follow-up. Within the CD4+ T cell population, the fraction of proliferating cells was significantly higher in patients before seed implantation, and remained almost unchanged with minor fluctuations throughout the whole follow-up period, and was restricted to CD4+Foxp3− Teff cells. We think this increased proliferation, especially in the later phases of the follow-up, was most probably the consequence of reduced CD4+ T cell pools, called homeostatic proliferation [42] rather than persistent T cell activation and clonal proliferation. This is supported also by the fact that the fraction of CTLA4-expressing CD4+ Teff cells was significantly lower in patients 24 and 36 months after treatment compared to both pre-treatment values and healthy controls. The expression of the CTLA4 activation marker is equally present on both CD4+ Teff cells and Tregs. However, while CTLA4 expression is induced on Teff cells in the initial phases of T cell activation, it is constitutively expressed on CD4+Foxp3+ effector Tregs. It plays an important role in immune suppression via regulating CD80/86 expression on DCs [43]. Another key immune activation marker present on both Teff cells and Tregs is PD1, which regulates the immune response at a later stage [44]. However, we did not test other direct CD4+ T cell activation markers, such as CD25 or CD62, therefore, clonal proliferation due to CD4+ T cell activation also cannot be excluded especially at earlier time points after RT. This is supported by data from Makoto et al., who reported that the fraction of activated CD4+ T cells increased in the blood of prostate cancer patients treated with LDR brachytherapy during a 15-month follow-up period [45]. 

Tregs are key players in the local and systemic tumor immune environments [46]. We investigated the long-term phenotypical changes in the Treg population identified through Foxp3 expressing CD4+ T cells. Various Treg subpopulations have been identified in humans based on the different cell surface and intracellular markers, out of which the expression of the Foxp3 transcription factor in CD4+ T cells represents one of the most frequently used ones. Though, it should be noted that a minor fraction of Foxp3 expressing CD4+ T cells (CD45RA-Foxp3^low^) do not have suppressive activity [47]. Prostate tumor infiltration with Tregs is an important indicator of an immune suppressing microenvironment and low tumor immunogenicity [48], elevated intratumoral Tregs in prostate cancer patients indicate poor prognosis and low survival rates [49]. Ashley et al. reported significantly elevated Treg levels both in the blood and in the tumor tissue of early-stage prostate cancer patients undergoing prostatectomy [50]. In our study Foxp3+ Tregs were not significantly different in the blood of prostate cancer patients either before or after LDR brachytherapy up to 24 months after RT. However, a significant 2.75-fold increase compared to pre-treatment values was measured in the Treg fraction in samples collected 36 months after RT. CTLA4 expression level on Treg cells, as well as the ratio of Foxp3+/Foxp3− cells within the CTLA4+CD4+ T cells were progressively increased after seed implantation reaching its maximum 36 months after therapy (Figure 3). This indicates that the fraction of Tregs with an effective immune suppressive phenotype was persistently higher in prostate cancer patients compared to healthy controls and had an increasing tendency during the 36-month follow-up. 

CD39 is expressed on Foxp3+ Tregs, where through its ATPase activity it is the rate limiting enzyme in generating the immune suppressive adenosine [51]. It is also expressed on CD4+ Teff cells, in which their presence is associated with apoptosis proneness [52]. While increased CD39 expression on Treg cells was mild and only transient, increased CD39 expression on Teff cells was much stronger; it was present in patients before and at early time points after RT and mildly increased CD39 levels persisted even 36 months later pointing to long-lasting damage in CD4+ Teff cells and potentially explaining the persistent depletion in the CD4 pool.

Altogether, reduced CD4+ T cell pools and their altered viability (increased CD39 expression on CD4+ Teff cells and a compensatory increase in their proliferative capacity) and an increase in the fraction of Tregs with suppressive phenotype (increased CTLA4 and PD1 expression) indicate the presence of a systemic immune suppression in prostate cancer patients before seed implantation affecting the CD4+ T cell arm of the adaptive immune system. The dynamic of changes in the investigated immune cell subpopulations at the time points after therapy where dose delivery was maximal (6–12 months) indicates that LDR brachytherapy had no substantial influence on immune cells with the exception of proliferating CD4+ Teff cells, which were transiently reduced 12 months after seed implantation. Therefore, we think that alterations to the adaptive immune response were long-lasting and progressed over time and were most probably a consequence of the initial malignant condition, although an additional immunomodulatory effect of chronic low dose radiation exposure cannot be excluded.

Next, we investigated systemic changes in the innate immune system by focusing on NK cells and DCs. Five different maturation stages of circulating NK cells can be distinguished within the CD3- lymphocytes according to the expression of CD16 and CD56 markers: immature precursor NK cells (CD56^bright^CD16-); immature/early mature precursor NK cells (CD56^bright^CD16^dim^), which are efficient cytokine producers and have immunoregulatory properties, but they become cytotoxic upon differentiation [53]. Mature, cytotoxic NK cells (CD56^dim^CD16^bright^) are the most frequent NK subpopulation in peripheral blood. Upon activation, their CD16 molecule is quickly shed from the cell surface, leading to degranulating NK cells with a CD56^dim^CD16- phenotype [53,54]. Anergic NK cells (CD56-CD16^bright^) constitute the second largest subset of CD3- lymphocytes after mature, cytotoxic NK cells in healthy individuals [55], and their level can increase in chronic infections and after persistent tumor ligand stimulation [56]. The fraction of total NK cells was significantly lower in the blood of prostate cancer patients before treatment but returned to control levels very rapidly. This was mainly due to changes in the mature cytotoxic NK cells as the main NK subpopulation, and a similar tendency was seen in the fraction of immature precursor as well as early mature NK cells. In contrast, the fraction of anergic NK cells increased in the tumor-bearing patients before and early after treatment but decreased as the tumor disappeared and remained stably below control levels. This indicates a phenotypical shift within NK cells towards a non-functional anergic subtype clearly linked with and probably also induced by the malignant condition. The fact that the malignant condition induces the ”education” of NK cells and increases the fraction of the anergic NK subpopulation has been associated with multiple malignant tumors [57,58]. LDR brachytherapy-related changes in the distribution of different NK cell subpopulations were transient and restricted to a mild increase in the fraction of early mature precursor NK cells and a reduction in the fraction of degranulating NK cells, which were most prominent 12 months after seed implantation when dose delivery reached its maximum. Altogether, these changes indicate that although prostate cancer was associated with a substantial redistribution of circulating NK cells in favor of a less functional anergic subpopulation; tumor regression led to the complete regeneration of the NK cell pool and its functional integrity. 

NKT-like cells are at the interface between the innate and adaptive immune system, they respond to glycolipids and stress-related proteins and are important players in tumor immune surveillance [59]. NKT-like cells share phenotypical and functional properties with both NK cells and T lymphocytes; they are capable of both quick cytokine secretion and direct target cell lysis and play an important role in tumor rejection [59,60,61]. NKT-like cell defects in the peripheral blood of prostate cancer patients were previously reported [62,63] and our results support these observations since we also found a significant decrease in the fraction of NKT-like cells in patients before treatment. 

Dendritic cells are the most efficient antigen-presenting cells and have an important role in anti-tumor immune responses [64]. DCs originate from CD34+ hematopoietic stem cells and can be classified into two main subpopulations: plasmacytoid and myeloid DCs. Data on cancer-related changes in the level of circulating DCs are contradictory. Minkov et al. found that both CD123+ lymphoid/plasmacytoid and CD11c+ myeloid DCs were important components of the anti-tumor immunity in non-small cell lung cancer (NSCLC) patients and their level could be a potential predictive factor for tumor development [65]. However, in another study, the level of both types of DCs significantly decreased in the circulation of cervical carcinoma patients [66]. Wilkinson et al. investigated three different types of DCs in the peripheral blood of prostate cancer patients (CD11c+CD16-, CD11c+CD16+, CD11c-CD123+) and did not find any significant differences in any DC subset, at any clinical stage of prostate cancer [67]. Our study showed an increased CD123+ lymphoid DC fraction in the peripheral blood of prostate cancer patients both before and after treatment throughout the whole follow-up period. Twelve months after seed implantation, when the delivered dose peaked, DC cell levels tended to normalize as a direct consequence of radiation exposure, but at later time points lymphoid DC levels went back to pre-treatment values. This very interesting pattern of the lymphoid DC pool kinetics might indicate an intrinsic defect in prostate cancer patients, and therefore, might constitute a predictive marker for tumor predisposition, thus supporting the findings of Minkov et al. in NSCLC patients. Our previous study investigating immune phenotype of RT-treated head and neck cancer patients also showed a strong increase in circulating DCs; however, in that study the follow-up period of patients after RT was very short (one month), thus no conclusions could be drawn on the long-term changes [29]. 

Multiple growth factors and inflammatory proteins have been linked with tumor stage, prognosis and metastasis proneness and their long-term follow-up might correlate with therapy-related side effects, such as fibrotic reactions or the development of second malignant neoplasms [68,69,70,71,72].

Endothelial dysfunction has been reported in cancer patients and linked with increased tumor metastasis, therapy resistance, or increased proneness to radiotherapy-induced fibrotic reactions [73,74,75]. We followed the temporal evolution of several growth factors involved in fibrosis and/or endothelial dysfunction, as well as cytokines, chemokines and other soluble proteins characteristic of chronic inflammation in patients before and up to 3 years after therapy. Similar to data reported by Tanji et al. [76], we found a relatively low number of proteins detectable and/or significantly altered from the tested panel in the blood of prostate cancer patients. 

The PDGF family consists of five different, biologically active isoforms and they have main roles in cell differentiation, proliferation, migration, survival and angiogenesis. Increased PDGF levels have been associated with malignant disorders [77,78]. Paracrine PDGF signaling triggers the epithelial-mesenchymal transition, thereby affecting tumor growth, angiogenesis, the formation of tumor blood vessels, invasion and metastasis [79,80]. Jiwen et al. found that PDGF-BB enhanced prostate cancer growth by promoting the proliferation of mesenchymal stem cells [81]. It was also shown that tumor cells were able to produce pro-angiogenic cytokines including VEGF, PDGF and FGF in response to ionizing radiation. These factors could protect endothelial cells and vessels from radiation-induced damage. Elevated levels of these factors correlated with negative clinical prognosis in different tumor types [82,83]. Our data show that PDGF AA and BB were the most strongly altered soluble proteins in prostate cancer patients before therapy but their levels rapidly returned to control values as the tumor was cured. RANTES (CCL5) is an important member of the CC subfamily of chemokines, which has an important role in promoting proliferation, angiogenesis, metastasis and drug resistance of prostate cancer cells via its crosslinking with chemokine (C-C motif) receptor 5 (CCR5) or CCR1. Its blood level positively correlated with poor prognosis in prostate cancer patients [84]. CXCL5 or *ENA78* is a member of the CXC chemokine family and has a role in the regulation of angiogenesis [85]. It was reported to increase cell migration and epithelial-to-mesenchymal transition of hormone-independent prostate cancer patients [86]. Similar to PDGF-AA and BB both CCL5 and CXCL5 plasma levels were significantly elevated in patients before therapy and rapidly normalized at later time points. These data indicate that increased levels of the four proteins in the blood were associated with the presence of the malignant tumor and thus they could be used as potential disease markers. The fact that no correlation with the pretreatment PSA values could be detected indicates that the investigated markers are not prostate-specific.

## 5. Conclusions

In the present paper, we followed systemic immunologic alterations in prostate cancer patients with a good prognosis, treated and cured with low dose rate brachytherapy. Patients were followed up for three years after the initiation of radiotherapy at regular intervals (three times in the first year and yearly thereafter). This long-term follow-up allowed us to identify both disease- and therapy-related as well as residual immunological damage. We showed that in prostate cancer patients before therapy both the adaptive immune system (altered CD4+ Teff cells and increased fraction of Tregs with suppressive phenotype) and the innate immune system (decreased NK cell fraction) were altered. However, while the innate immune response recovered as the tumor was cured, a mild long-term deficit in the adaptive immune response was observed, which persisted even 3 years later. An interesting finding was the constantly elevated fraction of lymphoid DCs in patients before treatment and throughout their follow-up, potentially suggesting that individuals with elevated lymphoid DC levels are at an increased risk for cancer development. Based on the kinetics of radiation energy deposition and the dynamics of changes in the investigated immune parameters we think that immunological changes that could be attributed directly to the effect of LDR brachytherapy were less characteristic. This might be due to the nature of the applied RT, since LDR brachytherapy leads to the deposition of radiation energy strictly locally over a long period of time, leading to very low dose chronic ionizing radiation exposure to the blood. As reported in previous studies, such low dose chronic exposures even in a total-body irradiation setting lead to very discrete alterations in systemic immune parameters, which in general remain within the limits of inter-individual variations within a population [87,88,89]. Although, in the absence of a suitable control group not treated with radiotherapy it cannot be evaluated with certainty which immune changes are due to radiation effects or due to the malignant condition itself. Within the literature there are very few reports on the long-term follow-up of the immune status of cancer survivors and the detected immune alterations were attributed to both cancer itself and treatment-related long-term consequences [90,91,92].

## Figures and Tables

**Figure 1 cancers-14-03058-f001:**
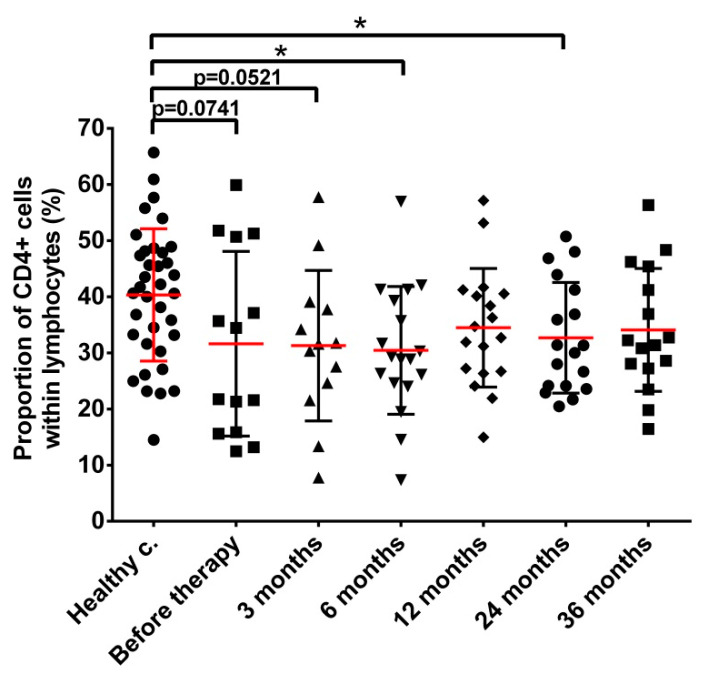
The fraction of CD4+ lymphocyte level is mildly decreased in prostate cancer patients throughout the whole follow-up period. Blood collection, processing and immune phenotyping were carried out as described in Materials and methods at the indicated time points relative to brachytherapy. *n* (healthy controls) = 36, *n* (cancer patients) = 13–18. Significant differences were indicated with * (*p* < 0.05).

**Figure 2 cancers-14-03058-f002:**
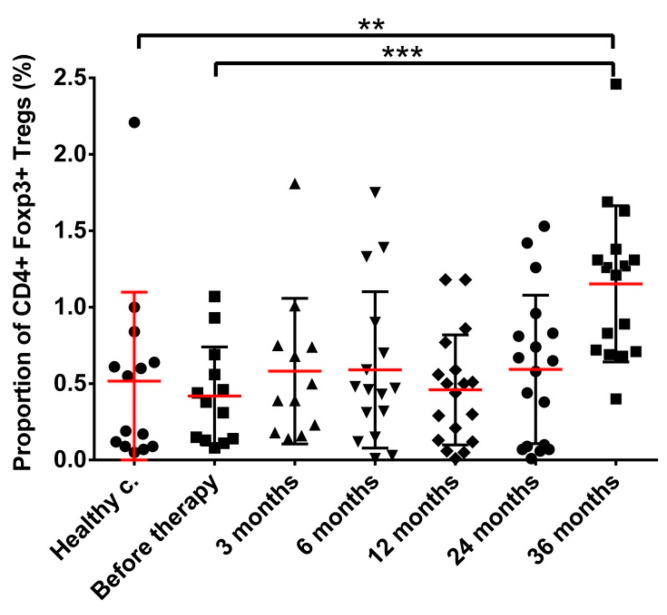
The proportion of Treg cells increases 36 months after radiation source implantation. Blood collection, processing and immune phenotyping were carried out as described in Materials and methods at the indicated time points relative to brachytherapy. *n* (healthy controls) = 14, *n* (cancer patients) = 12–18. Significant differences were indicated with ** (*p* < 0.01) and *** (*p* < 0.001).

**Figure 3 cancers-14-03058-f003:**
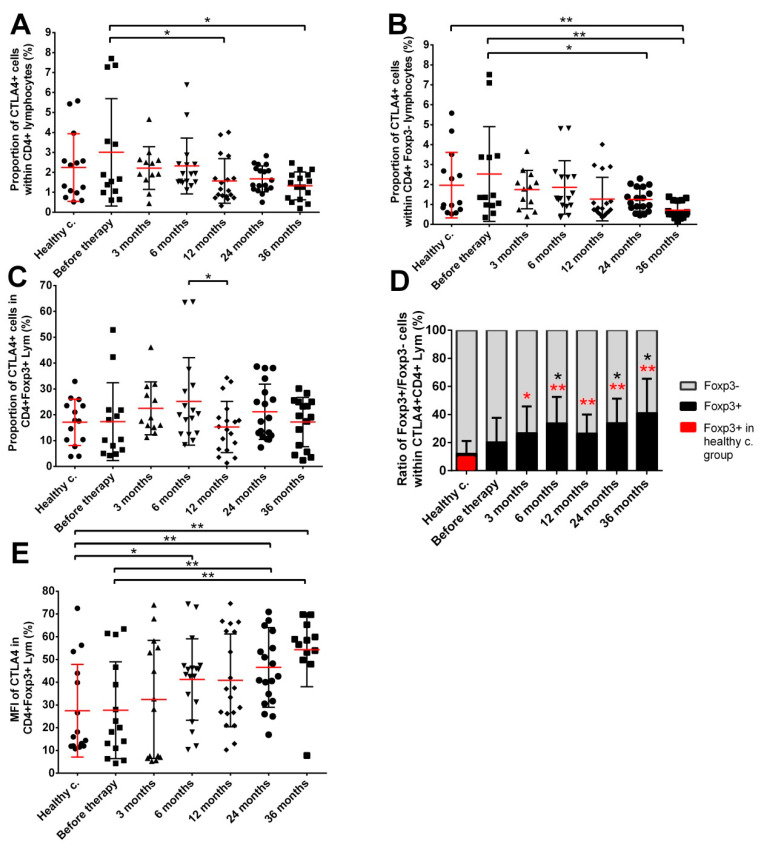
The fraction of Foxp3+ expressing CD4+CTLA4+ cells is progressively increasing in prostate cancer patients. (**A**) CTLA4 expressing CD4+ T cells; (**B**) CTLA4 expressing CD4+ effector T cells; (**C**) CTLA4 expressing Foxp3+ Tregs; (**D**) distribution of Foxp3+ and Foxp3− cells within the CTLA4+CD4+ lymphocytes. (**E**) CTLA4 expression level on CD4+Foxp3+ Tregs as determined by MFI. Blood collection, processing and immune phenotyping were carried out as described in the Materials and Methods at the indicated time points relative to brachytherapy. *n* (healthy controls) = 14, *n* (cancer patients) = 12–18. In Figure (**D**): red asterisks show significant changes compared to healthy controls and black asterisks represent significant changes compared to pre-treatment values. Significant differences were indicated with * (*p* < 0.05) and ** (*p* < 0.01). MFI: median fluorescent intensity.

**Figure 4 cancers-14-03058-f004:**
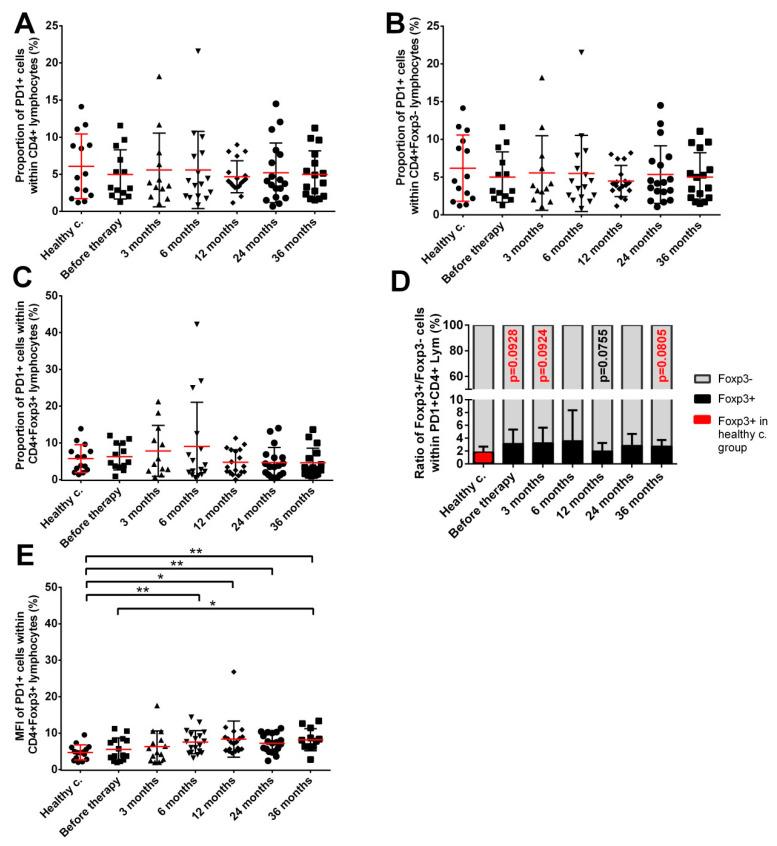
The fraction of Foxp3+ expressing CD4+PD1+ cells tends to increase progressively in prostate cancer patients. (**A**) PD1 expressing CD4+ T cells; (**B**) PD1 expressing CD4+Foxp3− effector T cells; (**C**) PD1 expressing Foxp3+ Tregs; (**D**) distribution of Foxp3+ and Foxp3− cells within the PD1+CD4+ lymphocytes; (**E**) PD1 expression level on CD4+Foxp3+ Tregs as determined by MFI. Blood collection, processing and immune phenotyping were carried out as described in the Materials and Methods at the indicated time points relative to brachytherapy. *n* (healthy controls) = 14, *n* (cancer patients) = 12–18. MFI: median fluorescent intensity. Significant differences were indicated with * (*p* < 0.05) and ** (*p* < 0.01).

**Figure 5 cancers-14-03058-f005:**
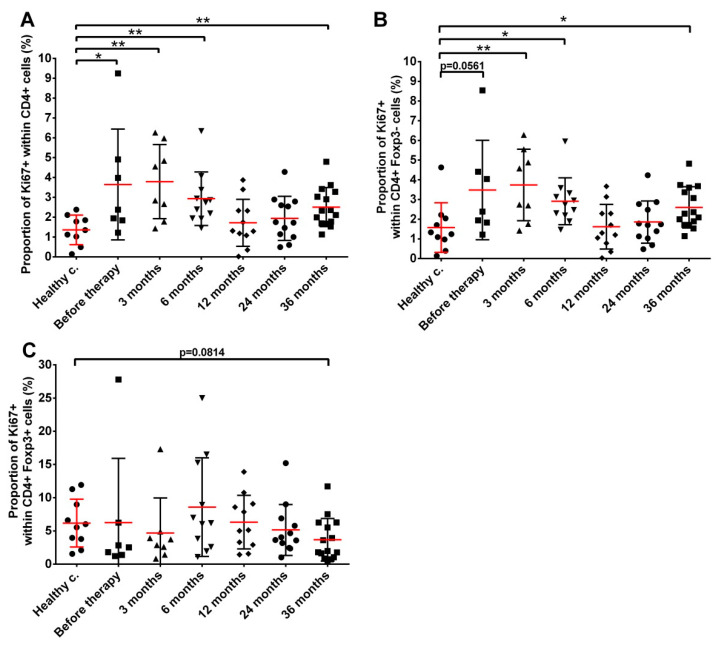
The fraction of proliferating CD4+ T cells was higher in prostate cancer patients compared to healthy controls. (**A**) Proliferating CD4+ cells; (**B**) Proliferating CD4+ effector T cells; (**C**) Proliferating Treg cells. Blood collection, processing and immune phenotyping were carried out as described in the Materials and Methods at the indicated time points relative to brachytherapy. *n* (healthy controls) = 10, *n* (cancer patients) = 8–18. Significant differences were indicated with * (*p* < 0.05) and ** (*p* < 0.01).

**Figure 6 cancers-14-03058-f006:**
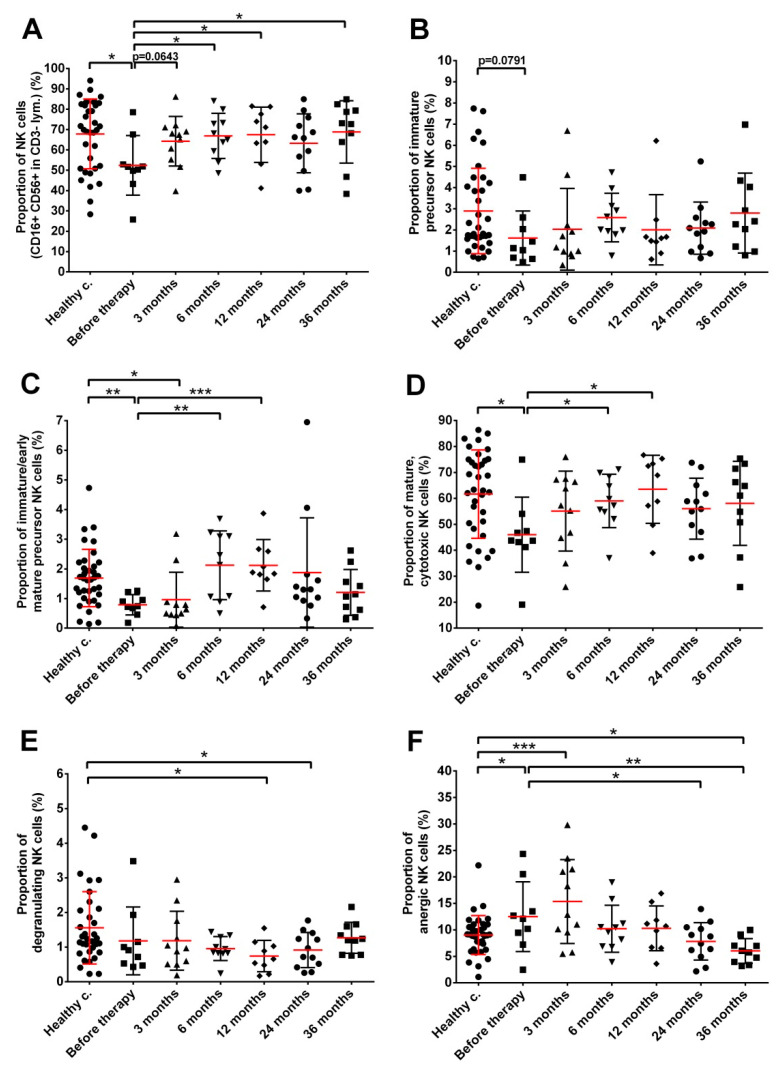
A pronounced redistribution occurs within the different NK cell subpopulations in prostate cancer patients treated with LDR brachytherapy. (**A**) circulating NK cells (CD16+CD56+CD3- lymphocytes), (**B**) immature precursor NK cells, (**C**) early mature NK cells, (**D**) mature, cytotoxic NK cells (**E**) degranulating NK cells levels, (**F**) anergic NK cells. Blood collection, processing and immune phenotyping were carried out as described in Materials and methods at the indicated time points relative to brachytherapy. *n* (healthy controls) = 10, *n* (cancer patients) = 8–18. Significant differences were indicated with * (*p* < 0.05) ** (*p* < 0.01) and *** (*p* < 0.001).

**Figure 7 cancers-14-03058-f007:**
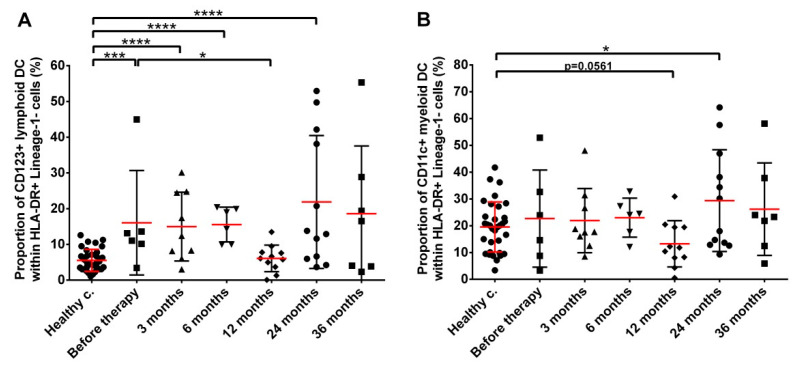
Persistent increase in the fraction of lymphoid DCs and milder changes in myeloid DCs in prostate cancer patients. (**A**) Lymphoid DCs, (**B**) Myeloid DCs. Blood collection, processing and immune phenotyping were carried out as described in the Materials and Methods at the indicated time points relative to brachytherapy. *n* (healthy controls) = 36, *n* (cancer patients) = 12. Significant differences were indicated with * (*p* < 0.05), *** (*p* < 0.001) and **** (*p* < 0.0001).

**Figure 8 cancers-14-03058-f008:**
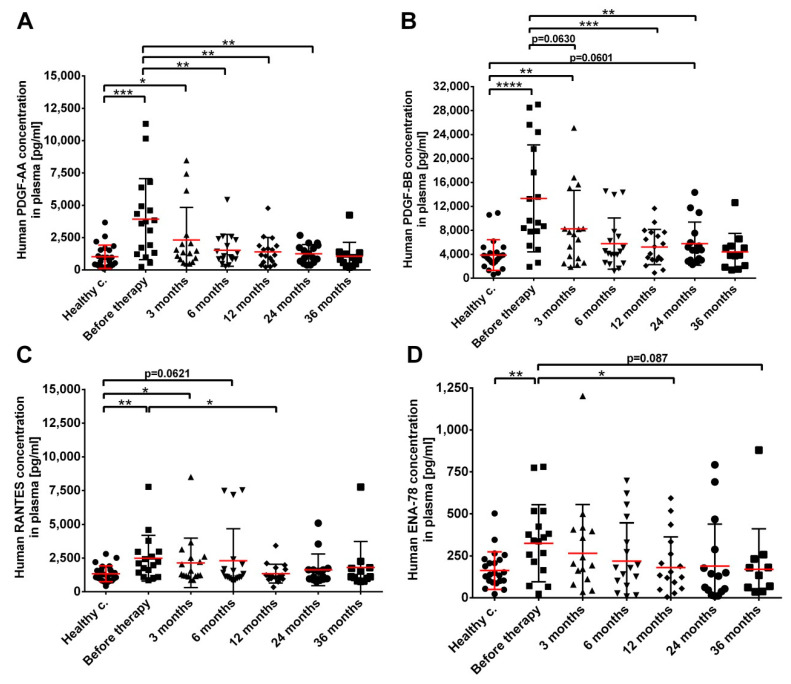
Increased (**A**) PDGF-AA, (**B**) PDGF-BB, (**C**) RANTES/CCL5 and (**D**) ENA-78/CXCL5 levels are associated with the presence of the tumor in prostate cancer patients. Blood collection, processing and plasma analysis was carried out as described in the Materials and Methods at the indicated time points relative to brachytherapy. *n* (healthy controls) = 24, *n* (cancer patients) = 12–18. Significant differences were indicated with * (*p* < 0.05) ** (*p* < 0.01) *** (*p* < 0.001) and **** (*p* < 0.0001).

**Table 1 cancers-14-03058-t001:** Demographics and clinical parameters of prostate cancer patients receiving low-dose rate brachytherapy. Risk classification scale is from 1 to 3. Low risk (1): PSA level before therapy < 10 ng/mL, GS 2-6 and T1-T2a; medium risk (2): PSA level before therapy = 10–20 ng/mL and/or GS 7 and/or T2b. Patient 94 had no Gleason score because of the unsuccessful biopsy. * patient 72 had duodenal ulcer, patient 85 had lymph node metastasis, patient 91 had distant metastasis.

Patient Code	Age (years)	Risk Classification	TNM Classification	Gleason Score (GS)	Hormone Type	Duration of Hormone Therapy Before RT (months)
P053	62	2	cT2c cN0 M0	3 + 2	0	0
P056	73	1	cT2a cN0 M0	3 + 3	LHRH antagonist	6
P057	61	1	cT2b cN0 M0	3 + 3	LHRH agonist	3
P058	67	2	cT2a cN0 M0	3 + 4	0	0
P061	73	2	cT2c cN0 M0	3 + 3	0	0
P062	69	2	cT2a cN0 M0	3 + 3	LHRH agonist	3
P068	72	2	cT1c cN0 M0	3 + 3	TAB	3
P069	62	1	cT2b cN0 M0	3 + 2	0	0
P071	74	3	cT1c cN0 M0	3 + 3	LHRH agonist	38
P072 *	68	2	cT2b cN0 M0	3 + 4	LHRH agonist	1
P080	75	2	cT1c cN0 M0	3 + 4	LHRH agonist	5
P081	76	1	cT2a cN0 M0	2 + 2	LHRH agonist	34
P083	72	2	cT2c cN0 M0	3 + 3	LHRH agonist	3
P084	63	1	cT2a cN0 M0	2 + 3	LHRH antagonist	3
P085 *	52	2	cT2c cN0 M0	3 + 3	0	0
P091 *	72	2	cT2c cN0 M0	3 + 3	LHRH agonist	3
P092	77	2	cT1c cN0 M0	3 + 4	LHRH agonist	3
P094	69	1	cT2a cN0 M0	-	LHRH agonist	6
P095	70	1	cT2a cN0 M0	3 + 3	0	0
P170	64	2	cT2c cN0 M0	3 + 3	0	0
P176	72	2	cT1c cN0 M0	3 + 3	0	0

**Table 2 cancers-14-03058-t002:** PSA values (ng/mL) of prostate cancer patients before and at multiple time points after brachytherapy. ‘*n*’: not available.

	PSA Level (ng/mL)					
Patient Code	Before Therapy	3 Months after Therapy	6 Months after Therapy	12 Months after Therapy	24 Months after Therapy	36 Months after Therapy
P053	7.00	0.57	0.28	0.01	0.14	0.10
P056	0.90	1.00	0.69	0.40	0.55	0.30
P057	1.64	2.07	1.36	1.89	1.40	0.40
P058	12.20	6.44	2.94	1.04	0.28	*n*
P061	11.40	1.87	1.07	0.01	0.21	0.16
P062	0.93	1.63	1.28	0.76	0.68	0.77
P068	0.09	0.60	0.90	0.60	0.60	0.40
P069	4.73	1.15	0.60	2,10	1.20	1.50
P071	1.14	0.50	0.34	0.59	0.23	1.34
P072	13.9	0.09	0.04	*n*	*n*	*n*
P080	0.09	0.30	0.33	0.42	1.60	5.70
P081	1.73	0.09	0.06	0.07	0.15	*n*
P083	0.75	0.38	0.41	1.13	0.43	0.26
P084	*n*	0.05	0.07	0.15	0.35	0.05
P085	5.01	5.50	6.60	0.15	*n*	*n*
P091	4.74	2.3	1.97	*n*	*n*	*n*
P092	0.09	0	0	0.16	0.27	0.11
P094	2.55	0.8	0.61	0.82	0.45	*n*
P095	8.13	0.70	0.57	0.27	0	*n*
P170	7.80	0.98	0.37	0.38	*n*	*n*
P176	12.4	1	0.09	0.06	*n*	*n*

## Data Availability

Not applicable.

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
