# Peer review of "Prostate Cancer Survivors Present Long-Term, Residual Systemic Immune Alterations"

_cancers, 2022, doi:10.3390/cancers14133058_

Round 1

Reviewer 1 Report

This study followed patients with low stage prostate cancer over 36 months and longitudinally evaluated the immune fraction as a result of RT.

  1. Methods -The timing of blood collection and flow cytometry is unclear. Were all of the samples run at the same time in one flow run? Adding clarity in methods will help improve this section for readers
  2. Supplemental figures with dot plots comparing healthy patients vs cancer patients will be helpful to readers. Having immune related changes for individual patients plotted over time could be added to supplemental section for further clarity
  3. Figures- Adding absolute values rather than proportion for immune fractions will be helpful to readers. This would be an important change as proportions can be misleading. Not sure what the value of figure 1 is, it could be moved to supplemental. Also please represent each patient as a dot to add more clarity
  4. Tables- adding the hormonal therapies administered to patients can be included to table 1 for more clarity
  5. An important control that is missing are prostate cancer patients that did not receive RT therapy. This would help dissect what changes are due to RT therapy and what changes were the result of malignancy as the results were compared to healthy controls.

Author Response

We thank the Reviewer for the thorough review of the manuscript and for the helpful comments. Please find below our detailed replies.

  1. Methods -The timing of blood collection and flow cytometry is unclear. Were all of the samples run at the same time in one flow run? Adding clarity in methods will help improve this section for readers.”

Blood collection was performed at every follow-up of the patients, namely before radiotherapy, 3, 6, 12, 24 and 36 months after the initiation of the brachytherapy. This was specified in the Methods section. PBMCs were selectively isolated by Histopaque gradient centrifugation and frozen in culture medium containing DMSO and stored in liquid nitrogen until assayed). The samples from multiple time points of each individual patient were investigated at the same time, in one flow cytometry assay. The manuscript was revised accordingly.

  1. Supplemental figures with dot plots comparing healthy patients vs cancer patients will be helpful to readers. Having immune related changes for individual patients plotted over time could be added to supplemental section for further clarity.

We thank the Reviewer for this suggestion. We included representative dot plot images (representing Treg and total NK cell populations) from healthy controls, cancer patients before and at certain time points after therapy initiation. For Treg subpopulations we also included graphs showing changes for individual patients (coloured individually) plotted over time. Please see Supplementary figures 4 and 6.

  1. Figures- Adding absolute values rather than proportion for immune fractions will be helpful to readers. This would be an important change as proportions can be misleading. Not sure what the value of figure 1 is, it could be moved to supplemental. Also please represent each patient as a dot to add more clarity.

Thank you for the valuable suggestion. We changed the graphs and now each patient is represented as a dot.

We moved Figure 1 into the supplement.

Regarding absolute values rather than proportions: we do not think absolute values are more accurate in our case. Please consider that immune phenotyping was not performed from fresh blood (in which case cell numbers represent absolute cellular distributions in the patients) but they were PBMCs isolated through a Histopaque gradient centrifugation. Since PBMCs are present in the layer between the Histopaque and the plasma, PBMC numbers greatly depend on layer thickness and the accuracy of removing this layer. Therefore, PBMC numbers retrieved after Histopaque selection do not reflect real PBMC cellularity of the patients, because certain white blood cell populations are removed by Histopaque, and because a fraction of PBMCs are lost during the isolation process itself. However, within the isolated PBMCs the fraction of the different lymphocyte subpopulations are representative for changes in lymphocyte subpopulations within the whole blood.

Nevertheless, we calculated absolute values of Tregs based on flow cytometry data. Treg numbers were calculated taking into account total PBMC numbers retrieved after Histopaque isolation. Numbers were normalized either to 10 million cells or to 1 ml blood (taking into account that 12 ml blood was subjected to Histopaque purification). Changes and the tendency of changes are very similar. Therefore, we decided to keep our graphs representing cell proportions instead of showing numerical values, but as an example, we included changes in the Treg population illustrated as cell fraction and as numerical changes in the supplementary material. Please accept our reply.

  1. Tables- adding the hormonal therapies administered to patients can be included to table 1 for more clarity.

We thank the Reviewer for this suggestion. The table was revised accordingly.

  1. An important control that is missing are prostate cancer patients that did not receive RT therapy. This would help dissect what changes are due to RT therapy and what changes were the result of malignancy as the results were compared to healthy controls.

Thank you for this comment, it is a very essential point. We agree, that in the absence of patients not treated with radiotherapy we cannot define for certainty which immune changes are due to radiation effects or due to the malignant condition itself.

This study constitutes part of a large study in which prostate cancer patients treated with four different radiotherapy modalities (low dose rate and high dose rate brachytherapy, teletherapy and ciberknife therapy) are followed in the long run to evaluate and compare radiotherapy-related late side effects between the different treatment protocols and investigate systemic immune changes. Since the major aim is to quantify radiotherapy-related side effects and see whether they can be correlated with systemic cytogenetic and immune changes, patients not treated with radiotherapy were not included.

This manuscript constitutes the first report of our results focusing on LDR brachytherapy patients. This treatment leads to mild late toxicities in terms of tissue reactions, developing less frequently than after other radiotherapy protocols, which was confirmed by our studies as well, still the extent of immune changes we found persisting 3 years after the start of the therapy was substantial and we considered it interesting for publication. Within the literature there are very few reports on long-term follow-up of the immune status of cancer survivors. Immune senescence and increased memory T cells were reported in testicular cancer survivors and in childhood cancer survivors (DOI: 10.3389/fonc.2020.564346, DOI: 10.1002/cam4.3788), altered cellular and humoral responses (including increased baseline Treg levels) to influenza vaccine were detected in various cancer survivors with chronic fatigue (DOI: 10.1080/21645515.2015.1040207). In all these cases long-term immune changes were attributed to both cancer itself and treatment-related long-term consequences. Within our study we present detailed analysis of delayed immunological changes detected in prostate cancer survivors. We investigated longitudinal changes in the functional integrity of CD4 cells by using multiple activation and proliferation markers, five different maturation stages of NK cells, different dendritic cells, all relevant for normal immune system functioning. For some immune parameters (eg. proliferating CD4 cells, certain NK subpopulations, lymphoid DCs) we think we can exclude with high probability that late changes are due to radiation. On the other hand, for other parameters (eg. Tregs, certain activation markers on CD4 cells) the effect of radiation on late immune changes cannot be excluded with certainty.  Even if our studies cannot dissect unambiguously therapy-related and malignancy-related changes, we think our studies help in elucidating long-term immune dysfunction in prostate cancer survivors and bring new, so far unpublished information within this topic.  We revised the Discussion and Conclusion section of the manuscript accordingly.

We kindly ask you to accept our replies.

Reviewer 2 Report

Overall, the study was well designed and executed however, there is some concern that should be addressed.

a. In several instances where the author checks the level of CTLA4, PD-1, and CD39, instead of proportion, MFI should be presented to reflect the expression level of these molecules on different lymphocytes.

b. Have the author checked the CD8 levels it would be interesting to look at those.

c. What is the response rate for this therapy? What was the response status in these cohorts? It would be interesting to look and see if there is any difference in immune parameters based on the response rate.

Author Response

We thank the Reviewer for the thorough revision of the manuscript and for the helpful comments.

Please find below our detailed replies to the comments.

  1. “In several instances where the author checks the level of CTLA4, PD-1, and CD39, instead of proportion, MFI should be presented to reflect the expression level of these molecules on different lymphocytes.”

We thank the Reviewer for this very helpful comment. We re-analysed our data regarding expression of CTLA-4, PD-1 and CD39 on Treg cells and indeed they strongly supported our observations that Treg suppressor activity is progressively increasing over time in prostate cancer survivors. We revised our manuscript and completed the Methods, Results and Discussion sections accordingly, as well as by adding new figures.

  1. Have the author checked the CD8 levels it would be interesting to look at those.

Unfortunately, we have not checked CD8 cells. We agree it would be interesting. Though, in this group of patients we decided to focus on innate immune system cells and on CD4 cells and their subpopulations, mainly Tregs and their activation status.

  1. What is the response rate for this therapy? What was the response status in these cohorts? It would be interesting to look and see if there is any difference in immune parameters based on the response rate.

Seed brachytherapy is usually applied for prostate cancer patients with good prognosis. In case of seed brachytherapy, the response rate evaluated right after the therapeutic intervention is not applicable, as the dose delivery is approximately one year long and the progression to relapse is slow in prostate cancer so we have to follow the patients for a minimum of 5 years, to gain a valid response rate. However, seed brachytherapy is known to result in excellent survival (15-year biochemical relapse-free survival was 80,4% in the study of Sylvester et al. and 85.9% in the study of D’Amico et al.). In our case, at five years, biochemical relapse-free survival is 89,9±6.8%, local relapse-free survival is 86,8±8.9%, and the overall survival is 90.5±6.4%. Due to the excellent response of patients to the therapy and the low relapse rates, it was not possible to compare immunological data with outcome variables.

We included this information in the Discussion of the revised manuscript.

Round 2

Reviewer 1 Report

All revisions have been reviewed and accepted.